# Host-Genome Similarity Characterizes the Adaption of SARS-CoV-2 to Humans

**DOI:** 10.3390/biom12070972

**Published:** 2022-07-12

**Authors:** Weitao Sun

**Affiliations:** 1School of Aerospace Engineering, Tsinghua University, Beijing 100084, China; sunwt@tsinghua.edu.cn; 2Zhou Pei-Yuan Center for Applied Mathematics, Tsinghua University, Beijing 100084, China

**Keywords:** SARS-CoV-2, host genome similarity, open reading frame, evade immunity

## Abstract

The severe acute respiratory syndrome coronavirus 2 (SARS-CoV-2) has a high mutation rate and many variants have emerged in the last 2 years, including Alpha, Beta, Delta, Gamma and Omicron. Studies showed that the host-genome similarity (HGS) of SARS-CoV-2 is higher than SARS-CoV and the HGS of open reading frame (ORF) in coronavirus genome is closely related to suppression of innate immunity. Many works have shown that ORF 6 and ORF 8 of SARS-CoV-2 play an important role in suppressing IFN-β signaling pathway in vivo. However, the relation between HGS and the adaption of SARS-CoV-2 variants is still not clear. This work investigates HGS of SARS-CoV-2 variants based on a dataset containing more than 40,000 viral genomes. The relation between HGS of viral ORFs and the suppression of antivirus response is studied. The results show that ORF 7b, ORF 6 and ORF 8 are the top 3 genes with the highest HGS. In the past 2 years, the HGS values of ORF 8 and ORF 7B of SARS-CoV-2 have increased greatly. A remarkable correlation is discovered between HGS and inhibition of antivirus response of immune system, which suggests that the similarity between coronavirus and host gnome may be an indicator of the suppression of innate immunity. Among the five variants (Alpha, Beta, Delta, Gamma and Omicron), Delta has the highest HGS and Omicron has the lowest HGS. This finding implies that the high HGS in Delta variant may indicate further suppression of host innate immunity. However, the relatively low HGS of Omicron is still a puzzle. By comparing the mutations in genomes of Alpha, Delta and Omicron variants, a commonly shared mutation ACT > ATT is identified in high-HGS strain populations. The high HGS mutations among the three variants are quite different. This finding strongly suggests that mutations in high HGS strains are different in different variants. Only a few common mutations survive, which may play important role in improving the adaptability of SARS-CoV-2. However, the mechanism for how the mutations help SARS-CoV-2 escape immunity is still unclear. HGS analysis is a new method to study virus–host interaction and may provide a way to understand the rapid mutation and adaption of SARS-CoV-2.

## 1. Introduction

Coronavirus disease 2019 (COVID-19) has been spreading globally for 3 years since December 2019, causing 543,323,503 confirmed cases and 6,328,552 deaths worldwide as of 29 June 2022. The severe acute respiratory syndrome coronavirus (SARS-CoV-2), a single-stranded positive-sense RNA virus, was reported as the cause of COVID-19. The SARS-CoV-2 has a linear genome of about 30,000 nucleotides [1,2]. Study shows that the sequence homology of SARS-CoV-2 and SARS-CoV is nearly 77% [1]. Although symptoms of SARS-CoV-2 infections are similar to SARS-CoV, epidemiological differences are apparent between the two diseases. Compared with the SARS in 2003, the ongoing COVID-19 pandemic appears to be more contagious [3]. The genomic basis for why this novel coronavirus is so widespread remains unclear.

The RNA virus is susceptible to gene recombination and the viral genome may contain traces of evolutionary history of infecting hosts, which can be identified by comparing genome similarity between the virus and the hosts. Although the genomes of viruses and their hosts are very different in general, nucleotide sequence similarities do exist. Such similarities have deep biological reasons. The genetic similarities may be inherited from a common ancestor and remain stable for their biological significance. It is also possible that similar pieces of genome happen to be preserved in viruses and hosts over time due to the biological benefits of gene products. In addition, when the virus interacts with the host, the exchange of genes between virus and the host may lead to genomic similarities.

More and more studies have reported on the similarity of virus and host gene sequences. Chang et al. [4] reported that five out of six samples could be amplified by Epstein–Barr virus (EBV)- or hepatitis C virus (HCV)-specific primers when using human peripheral blood DNA as template for polymerase chain reaction (PCR). Therefore, it is speculated that some genes of the two viruses may also exist in the human genome, or have homology with human genes.

Senkevich et al. [5] found some genes similar to specific segments of human genome in molluscum contagiosum virus (MCV). MCV is a human poxvirus that lacks genes associated with virus–host interactions in other poxvirus species (variola virus). However, a portion of the MCV gene that is highly similar to specific sections of the human genome is also hard to find in other pox viruses. These results suggested that these host-like genes may provide specific strategies for coexistence of MCV and host [5]. In other words, viruses are likely to use host-specific genes for activities related to virus–host interactions, such as evasion of host innate immunity.

Simian Virus 40 (SV40), the first animal virus to complete full-sequence DNA analysis, can infect monkeys and humans and cause tumors [6,7,8]. Rosenberg et al. [9] found that some mutant SV40 viruses contained nucleic acid sequences from their host monkeys. The findings suggest that this virus can recombine with host genes to complete its own physiological process. Such genomic recombination makes up for its lack of function or increases virulence. Shackelton et al. [10] believed that the evolution of large DNA viruses was a combination of viral genes themselves and host genes. The results suggested that the evolution of viral genes involves gene transfer between genomes and gene replication within genomes.

Selection pressure from host immune response plays an important role in virus mutation. Homology between virus and host proteins indicates the presence of host gene capture. A recent study has shown that the evolution of human genome was influenced by viral infections [11]. In mammals, nearly 30% of the adaptive amino acid changes in the human proteome are caused by viruses, suggesting that viruses are one of the main driving forces in both mammalian and human proteome [12]. Many viruses have evolved proteins that regulate or inhibit the host’s immune system by acquiring immune modulation genes from cells [13,14,15,16,17]. These studies support the possibility that viruses may exchange genetic information with host cells when they infect them. From an evolutionary point of view, natural selection can pick out traits that are easier to preserve. It can be inferred that most of the traits and mechanisms retained in “coevolution” between viruses and their hosts, including genetic and mutational mechanisms, benefit at least one or both. At the molecular level of evolution, virus–host mutual adaptation requires the exchange of genetic information. This may be responsible for the similarity of gene expression products and ultimately nucleotide sequences. In summary, the process of viral infection may lead to the subsequent use of genes from the other side by the virus and host, which represents a mutually beneficial aspect of virus–host interaction.

Coronavirus may also undergo gene exchange with human hosts. Such gene exchange can be detected from virus–host genome similarity. It is interesting to study the relationship between host-genome similarity (HGS) and viral transmission/pathological ability caused by gene exchange. The single-stranded RNA of coronavirus generally encodes three categories of proteins: (1) replica proteins ORF 1a and ORF 1ab, (2) structural proteins including S (spike), E (envelope), M (membrane), and N (Nucleocapsid), and (3) accessory proteins with unknown homologues. The structural protein genes are organized as ‘-S-E-M-N-’ in SARS-CoV-2 genome, and accessory protein genes are distributed between S and E, M and N. The glycoprotein S is crucial to virus binding and entry.

In general, accessory protein genes of coronavirus are believed not essential for virus replication in vitro [18,19,20,21,22,23,24]. However, as more and more evidence emerges, many accessory genes show important activity in virus–host interactions in vivo, such as regulating the host immune response [25,26,27,28,29,30,31,32,33,34,35,36,37]. Recent studies show that ORF 6 contributes in SARS-CoV replication and pathogenicity [30,38]. ORF 6 also suppresses the induction of IFN and the signaling pathways [25], which strongly suggests that ORF 6 plays a critical role in subverting host’s immune response during SARS-CoV infection. Evidence suggests that ORF 6 plays an important role in replication and pathogenicity of SARS-CoV [38,39,40,41,42]. An interesting study shows that overexpression of ORF 6 can induce DNA synthesis in mammalian cell [43].

An intact gene ORF 8 encodes a single accessory protein at the early stage of the SARS epidemic. Strikingly, a 29-nt deletion appears in most isolated SARS strains at later stage, which splits the gene ORF 8 into two fragments, ORF 8a and ORF 8b [44]. Although the linking of ORF 8a and ORF 8b into a single continuous gene fragment had no significant effect on virus growth and RNA replication in vitro [20], it is highly possible that the split in the gene ORF8 has connection with evolutionary adaption of SARS-CoV to human in vivo. Amazingly, studies show that SARS-related CoVs in horseshoe bats have 95% genome identities to human and civet SARSr-CoVs, but the ORF 8 protein amino acid similarities vary from 32% to 81% [45]. These findings indicate that the ORF 8 gene is more prone to mutation in virus–host interactions.

Accessory proteins ORF 8a and 8b have been observed in most SARS-infected cells [46]. Wong et al. [47] found that proteins ORF 8b and ORF 8ab in SARS-CoV have functions of inhibiting the IFN response in viral infection. Like ORF 6, overexpression of ORF 8b can induce cell DNA synthesis and inhibit the expression of viral envelope proteins [30]. It is also reported that ORF 8b can form insoluble intracellular aggregates and triggers cell death [48]. However, the manner in which ORF 8 modulates virus–host interaction and virus replication is still unknown.

The accessory protein genes play a key role in inhibiting innate immune response in vivo and are more susceptible to species-specific mutations under the pressure of evolutionary selection. Once inside the cell, the virus immediately confronts other critical proteins, unknown as the host-restriction-factors (HRF) [49]. HRF are proteins that recognize and block viral replication. Virus–host interaction controls the species-specificity and viral infection ability. Under the pressure from the host immune system, viruses must be able to cheat a range of constraints associated with the host species and often show evolutionary mutation selections. We hypothesized that accessory ORFs may retain beneficial mutations to increase high host-genome similarity (HGS). Identifying emerging genetic mutations in virus populations with high HGS may help to understand how SARS-CoV-2 evolved to adapt to humans. To my knowledge, there are few studies on the genetic similarities between SARS-CoV-2 and hosts.

A recent study shows that the Omicron variant is more infectious (about 2.7–3.7 times) than Delta among vaccinated people, but Omicron and Delta have nearly the same rate of infection in unvaccinated people. The results indicate that the vaccine’s blocking mechanism against the Delta strain appears to be less effective against Omicron. The increased transmissibility of the Omicron variant is closely related to its ability to evade immunity from vaccines, but how exactly Omicron evades immunity remains unclear.

This study aims to investigate the genomic similarity of SARS-CoV-2 to humans. It is the first attempt to understand links between infectivity/virulence and virus-host genome similarity. A quantitative definition of host-genome similarity (HGS) is proposed. A remarkable correlation is discovered between HGS and inhibition of antivirus response of innate immunity. ORFs with higher HGS can suppress the expression of IFN-β, IFN-I, ISRE and NFκB to a greater extent. Analysis of 40,016 SARS-CoV-2 genomes showed that that the HGS of ORF 7b and ORF 8 genes increased significantly in the past 2 years. In order to understand how genetic mutation accelerates SARS-CoV-2 adaptation, the relationship between mutations and HGS was analyzed for Alpha, Delta, and Omicron variants. The results revealed that there were a few mutations that survived in SARS-CoV-2 strains with high HGS. The mutation ACT > ATT is commonly shared by strains of Alpha, Delta, and Omicron variants, while other mutations are different in strains of the three variants. This finding provides strong evidence that only a very few mutations survived selection events and resulted in a new population of SARS-CoV-2 with high HGS. However, the mechanism of how mutations make SARS-CoV-2 more adaptable to humans remains unclear.

## 2. Materials and Methods

### 2.1. SARS-CoV-2 Genomes

The number of SARS-CoV-2 genomes in GISAID database [50,51] reached 10,736,775 as of 5 May 2022. The complete and high-coverage viral genomes with patient status and complete collection date reached 219,667. This study used a dataset of 40,016 high-quality SARS-CoV-2 genomes with geolocations such as China, the USA, and Europe. Sequences containing nucleotide names other than A, G, C, and T were discarded. Genomes with inconsistent ORF length were removed to maintain comparability of sequence similarity. The number of genomes for SARS-CoV-2 variants Alpha, Beta, Delta, Gamma, and Omicron are 15,284, 172, 23,477, 955, and 128, respectively, in the dataset.

Figure 1 shows the open reading frames (ORF) organization in SARS-CoV-2 (GenBank: MN908947.3) and SARS-CoV (GenBank: AY394850.2). Different nomenclature has been used in the literature [18,51,52], such as ORF6 in Narayanan et al. [52], ORF 7 in Marra et al. [53], and X3 in Rota et al. [18]. The coronavirus ORFs encode structural proteins such as spike (S), envelope (E), membrane (M), and nucleocapsid (N). In addition, some coronavirus ORFs (ORF 3a, 3b, 6, 7a, 7b, 8a, 8b, and 9b for SARS-CoV) can also encode accessory proteins, which are usually related to virus–host interaction in vivo. Such ORFs have high levels of mutations. However, the functions of accessory proteins are still not fully understood.

### 2.2. Host-Genome-Similarity (HGS) Definition

A quantitative definition of HGS is proposed to investigate the matching of viral ORF and human genome. The HGS represents the degree of sequence pattern similarity shared between virus and host genome. Karlin and Altschul studied statistical significance of sequence patterns [54]. Matching score has been defined to reflect nucleotide similarity and distinguish biologically relevant patterns. HGS is a further version of the matching score used to compare the virus–host gene similarity, especially for genome segments of different length.

The Basic Local Alignment Search Tool (BLAST) [55] is a family of bioinformatics algorithms for comparing gene and protein sequences. BLASTn, a member of BLAST program, is used to find local similarity between viral genome and human genomic database (*Homo sapiens* GRCh38.p12 chromosomes). A matching score *S* is produced by BLASTn to represent the statistical significance of similarity shared by virus and human genomes. A higher score means the sequences share significant Homologies.


(1)
$$H=\frac{1}{2n}\sum S,$$


The host-genome similarity (HGS) is defined as where n represents the length of the target sequence. The HGS value H is the ratio of the number of matched base pairs to the total length of the sequence when the matched sequences are converted into sequences of the same length.

Each of the viral ORFs has a HGS value, which represents the sequence similarity to host genome. The HGS of whole viral genome is calculated by a weighted sum of all the ORF HGS values. The weighting factor is the length ratio between the ORF subsequence and the entire viral genome. The ORF lengths of SARS-CoV and SARS-CoV-2 genomes are given in Table 1.

Based on the proposed model, HGS value of a virus RNA fragment is calculated by following steps:Prepare the subgenomic CDS sequence (such as the sequence of ORF 6) of virus strain.Choose human genomic database (*Homo sapiens* GRCh38.p12 chromosomes) database.Set BLASTn program parameters. The expect threshold as 10, word size as 11, penalty for a nucleotide mismatch as −3, reward for a nucleotide match as 2, gap costs are chosen as existence = 5 and extension = 2, max matches in a query range as 1.Run the BLASTn matching analysis.The HGS value is obtained through the matching scores produced by BLASTn analysis and the length of target sequence.

The parameters used in BLASTn have influence in matching virus and host genomes. The default parameters are used for the expected threshold, word size, and gap costs. Penalty for a nucleotide mismatch is −3 and reward for a nucleotide match is 2. Thus a reward/penalty (absolute) ratio of 0.66 (2/−3) is used for matching virus and host genome, which is appropriate for sequences with low conserved nucleotide patterns. The max matches in a query range is 1, which indicates that if the virus genome segment matches the host genome (no matter how many times it matches), it only reports once.

## 3. Results and Discussions

### 3.1. SARS-CoV-2 Delta Variant Shares More Genomic Similarity to Human

A previous study found that SARS-CoV-2 HGS was significantly elevated compared to the SARS-CoV [56]. In this work, the HGS values of viral ORF were calculated for genomes in two datasets. The ORF HGS is obtained by aligning viral subsequence with human genome (*Homo sapiens* GRCh38.p12 chromosomes). The first dataset contains 2594 genomes up to 20 May 2020 [57]. The second dataset contains 40,016 genomes up to 5 May 2022. Both datasets contain high-quality SARS-CoV-2 genomes with geolocations such as China, the USA, and Europe. The second dataset contains genomes of five SARS-CoV-2 variants. The number of genomes for Alpha, Beta, Delta, Gamma, and Omicron variants are 15,284, 172, 23,477, 955, and 128 respectively.

SARS-CoV-2 virus has undergone a series of important mutations in the past 2 years. The Delta and Omicron variants caused a global outbreak of COVID-19. In order to investigate whether the HGS changes are related to their transmissibility, HGS calculation has been performed for different variants. Amazingly, the results show that SARS-CoV-2 has a remarkable HGS increase in ORF 7b and ORF 8 over the last 2 years (Figure 2). The average HGS of ORF7b and ORF8 increased to 114% and 110% from 20 May 2020 to 5 May 2022, respectively.

Figure 3 shows that the SARS-CoV-2 variants display quite different HGS for the 10 ORFs. For all the SARS-CoV-2 strains, ORF 7b, ORF 6, and ORF 8 are the top three genes with the high HGS (Figure 3). The Delta variant has the largest HGS in ORF 7b, ORF 8 and ORF 6, while the Omicron variant has the lowest HGS values among all the viral variants (Figure 3).

Although the results show that the HGS of SARS-CoV-2 has increased during the last 2 years, many of the important details are yet to be uncovered. The SARS-CoV-2 variants may have had quite different HGS variations during the last 2 years. Data from all 40,016 virus strains were analyzed to obtain a more detailed picture of changes in HGS in the SARS-CoV-2 genome. In particular, changes in HGS over time are of concern. Figure 4 shows the HGS of ORF 8 gene vs. time for five different variants with a geolocation of Europe (35,875 genomes). Amazingly, the HGS distribution has a special pattern, forming five distinct horizontal lines. The data clearly demonstrate that the Delta variant overwhelmed other variants since mid-2021. The HGS data points of Beta and Gamma are basically concentrated on the third and fourth lines (from low H to high H). The HGS data points of the Omicron variant are mainly at the third line from low to high. Omicron variant has HGS values comparable to the average level of the Alpha variant, but much lower than Delta. However, due to small amount of data, detailed analysis of Omicron variants remains to be completed.

It would be more interesting if the same ORF has mutated with different HGS Figure 5 shows HGS vs. mutation variation in ORF8 gene of Delta variant. The dataset contains 3854 genomes with geolocation as Europe. The top 100 mutations with the highest frequency and their HGS values are shown in the figure. Figure 5 shows that the mutations almost cover the whole ORF8. Meanwhile, the HGS also shows a discrete pattern, forming three horizontal lines. The top three most frequent mutations were ACT > ATT, GAT > GGT, and GCT > GTT. They occurred 5215, 4116, and 3332 times in all 3854 genomes, respectively. Some mutations have significantly higher HGS. Mutations exceeding 90% of maximum HGS value are shown in Figure 5 These high HGS mutations include CCT > CTT, GTG > GTT, ATC > ATT, TCT > TTT, GGT > GTT and AGA > ATA. It should be noted that the HGS of the top three mutations with the most frequent occurrence in ORF8 gene do not exceed 90% of the maximal HGS value. These mutations, which occur most frequently, have relatively low HGS and may not have an advantage in evading the immune system.

Figure 6 shows the HGS of S gene (encoding spike protein) vs. time. The dataset contains 35,875 genomes of five different variants with geolocation as Europe. For S gene, the HGS of Delta variant is significantly lower than that of other variants (Figure 6). The HGS of Omicron variant is at the average level of the Delta. The relatively low HGS of Omicron is still a puzzle. More Omicron variant data may be needed in further studies.

Figure 7 shows HGS vs. mutation variation in S gene of Delta variant. The mutations also cover all the range of S gene. The HGS data points shows a discrete pattern forming five horizontal lines. Mutations exceeding 90% of maximal HGS value are shown in Figure 7 (pentagram). These high HGS mutations include ACT > ATT, GAT > GGT, GCT > GTT, AAC > AAT, GTG > GTT, ACT > ACA, TCC > TCT, ACC > ACT, GAC > CAC, TAC > CAC, and ACT > AAT. Surprisingly, the top three mutations with the most frequent occurrence in S gene all have very high HGS, exceeding 90% of the maximal HGS value of all mutations. This is different from ORF 8. The reason why the most common mutations in the S gene have such high HGS is not clear. Based on the findings in following section that HGS is correlated with the suppression of antiviral response of host innate immune, the most frequent mutations in the S gene may enhance the virus’s ability to evade the immune system.

In the ongoing SARS-CoV-2 pandemic, the number of infection cases by Omicron is growing much faster than Delta variant. The Omicron is highly transmissible and is about 2.7–3.7 times more infectious than Delta [58]. It is known that Omicron has 15 mutations in the receptor-binding domain (RBD) of its spike protein and can bind to human ACE2 more efficiently. Studies showed that Delta and Beta RBD mutations are confined with class 1 and 2 antibody epitopes [59]. Experiments showed that Beta escape from class 1 and 2 antibodies and Delta escape from class 2 antibodies [60,61]. In contrast, Omicron can interact with four antibody classes, suggesting an enhanced antibody escape breadth [59]. Recent studies showed that Omicron may be less severe than other coronavirus variants, but risk of reinfection with the Omicron variant is higher [62,63]. Why the Omicron is so different from other variants is still not clear, but viral genome mutation is believed as an important factor, especially in gene encoding spike protein.

### 3.2. Virus Genes Related in Suppressing Innate Immune Tend to Have High HGS

The viral genome and proteins of SARS-CoV have been studied in depth in the past decade, but the knowledge on SARS-CoV-2 accessory proteins is still limited. Coronavirus has evolved to escape the innate immune (especially IFN-I expression and signaling) through suppression of IFN induction and singling pathways by non-structural proteins (nsps), structural proteins (S, E, M, N), and accessory proteins (ORF 3a, 6, 7a, 7b, 8a, 8b) [31,32,33,34,35,36,37,64,65,66,67]. By introducing the concept of genome similarity of coronavirus to host, the mechanism of the rapid spread of the newly emerged SARS-CoV-2 variants may be inferred from the changes in HGS.

The ORF 7b, ORF 6, and ORF 8 are the top three genes with high HGS in SARS-CoV-2. The results imply that these genes may be essential to antagonistic relationship between virus and host, such as suppressing immune response. The ORF 1ab genes, which encodes two large RNA-dependent RNA polymerase and form the functional fragments of transcribing complementary RNA, are closely related to viral translation/replication process.

In order to understand the relation between HGS and immunomodulation at the genetic level, the relationship between the expression of immune tags in cells infected with viral proteins and the corresponding HGS changes is analyzed. A quantitative relationship between viral HGS and the strength of suppressing host immune system is established.

The adaption to host cell and ability to evade innate immune is crucial to rapid replication. The HGS of ORF 7b and ORF 8 in SARS-CoV-2 increased to 114% and 110% in the last 2 years. Such significant increments imply the improvement of the similarity between the virus and the host genome, which may be a reflection of increased adaption to humans. Studies show that ORF 3b, ORF 6, and N proteins of SARS-CoV enhance the ability to suppress the expression of IFN-β of host innate immunity [25]. When interferon binds to the cell receptor IFNAR, the JAK/STAT signaling pathway is activated, leading to activation of the IFN stimulated genes (ISGs) containing IFN-stimulated response element (ISRE) in the promoter. Expression of genes with ISRE will trigger the production of hundreds of antiviral proteins, which inhibit viral infections. So, a reduction in expression from the ISRE promoter is a direct indicator of the enhanced ability to inhibit interferon synthesis.

To quantitatively analyze the effect of these viral proteins on inhibiting interferon signaling and synthesis, the data on expression of ISRE promoter in cells transfected with SARS-CoV were analyzed. In the work of Kopecky-Bromberg et al. [25], the cells were transfected with plasmids expressing individual SARS-CoV proteins (ORF 3b, ORF 6 and N) and then infected with Sendai virus or treated with IFN-β 24 h after transfection. Both interferon synthesis and signaling are required to express ISRE promoter if cells are infected by Sendai virus. However, in IFN-β-treated cells, the expression of the ISRE promoter requires only interferon signaling. Cells transfected with empty vector plasmid are used as a negative control. The expression of ISRE promoter are percentages of the value for the empty control. We calculated the HGS of ORF 3b, ORF 6, and N for SARS-CoV.

In cells treated with IFN-β, Kopecky-Bromberg et al. [25] found that N protein did not significantly inhibit the expression of ISRE promoter. The expression level is about 78% of the value for the empty control. However, ORF 3b and ORF 6 still inhibit the expression of ISRE promoter. Here, for 293T cells transfected with the SARS-CoV proteins and infected by Sendai virus at 24 h posttransfection [25]. Amazingly, it is found for the first time that the expression of ISRE promoter decreased rapidly with the increase of HGS for the cells infected with Sendai virus (Figure 8).

The results showed that a correlation exists between the inhibition of interferon by coronavirus protein and its HGS. If this is true, one would expect to find the same phenomenon in SARS-CoV-2. A couple of recent studies have identified SARS-CoV-2 proteins that antagonize key components for antiviral response of host innate immune, such as type I interferon (IFN-β), NF-κB-responsive promoter and interferon-stimulated response element (ISRE) [68,69]. The experiments found that viral ORF 6, ORF 8, and nucleocapsid proteins were potential inhibitors of IFN-β signaling pathway. Additionally, only ORF 6 and ORF 8 proteins were able to inhibit the ISRE [68]. SARS-CoV-2 nonstructural protein 1 and 6 suppress IFN-I signaling more efficiently than SARS-CoV [69].

The relation between IFN inhibition of ORF proteins and HGS is examined here (Figure 9) The relative activity of promoter was normalized with the treated empty vector control value was set to 100%. Remarkably, there is a clear correlation between the suppression of IFN by viral proteins and HGS. The antiviral promoter activity of immune system decreased with the increase of HGS. These findings confirmed that HGS, i.e., similarity between virus and host genome, is a reliable indicator of the suppression of innate immunity by viral proteins.

### 3.3. Suppression of IFN-I Signaling by ORFs with Different HGS Values

Considering the significant HGS increments of ORF 6 and ORF 8, it is speculated that SARS-CoV-2 would further suppress the IFN I synthesis and delay host innate immunity. Channappanavar et al. shows that the rapid SARS-CoV replication and relative delay in IFN-I signaling result in immune dysregulation and severe disease in infected mice [70]. The deadly effects of SARS-CoV on human tissues and organs stem from uncontrolled inflammation response [71]. Enhanced suppression of immune response and further delayed type I interferon (IFN-I) signaling caused by increased host-genome similarity may account for the epidemiological manifestations of SARS-CoV-2, such as longer incubation periods, mild symptoms, rapid spread, and low mortality. However, even if the role of delayed IFN-I signaling in SARS infection has been confirmed, the mechanism of how SARS-CoV-2 causes further delay of immune signaling and how it leads to new immunopathological features remain largely unknown.

When virus enters cell, innate immunity is firstly activated to resist the replication process of viral genome. Virus has many genomic characteristics different from host cells, which are called pathogen-associated molecular patterns (PAMPs) [72]. An mRNA with a cap structure at the 5′-end is considered the self RNA of host cell. RNA with an uncapped 5′ triphosphate terminus or with a non-methylated/partially methylated cap structure are considered viral PAMPs. In addition, the virus produces double-stranded RNA as an intermediate during replication, which is also a prominent viral PAMP.

Host cells recognize non-self RNA species primarily through the three types of pattern recognition receptors (PRRs) [72]: Toll-like receptors (TLRs), retinoic acid-inducible gene I (RIG-I)-like receptors (RLRs), nucleotide-binding oligomerization domain (NOD)-like receptors (NLRs) [31,73]. Detection of PAMPs by the host cell PRRs leads to recruitment, phosphorylation, and dimerization of a range of downstream regulatory proteins. The activated protein complexes, including the interferon regulatory factor 3 and 7 (IRF3 and IRF7), then move into the nucleus and ultimately trigger expression of interferons (IFNs) and cytokines [74] (Figure 10).

For coronavirus with RNA genome, TLR3 and TLR7 are the two most important TLR receptors which induce innate immune response through the MyD88 and TRIF signaling pathways [31,37,75,76,77,78,79,80,81,82,83]. Figure 10 shows the signaling networks from detecting virus RNA to secreting type I IFN. The intracellular cytosolic RLRs consists two important members: retinoic acid inducible gene I (RIG-I) and melanoma-differentiation-associated protein 5 (MDA5). RIG-I mainly detect short RNA with 5′ triphosphate [84,85,86]. MDA5 recognizes large size RNAs and RNAs with cap structures lacking ribose 2′-O-methylation and long dsRNAs [87,88,89,90]. The produced IFNα/β are released out of cell and bind to the interferon-α/β receptor (IFNAR) located at cell surface (Figure 10), helping to establish an antiviral state in threatened cells.

Studies show that SARS-CoV did not induce a significant IFN response in host cells, suggesting that coronavirus may have evolved an effective way to evade innate immune [91,92]. Expression of IFNα/β are suppressed by SARS-CoV viral proteins, including but not limited to nucleocapsid (N) protein, ORF 3b, ORF 6, and many non-structural proteins (nsps) encoded by ORF 1a and ORF 1b [25]. It has been shown that ORFs of virus play important roles in evading innate immune by inhibiting antivirus productions and blocking signaling pathways at different stages [32,33,34,35,37,64,65,66,67].

Although the functions of accessory proteins of SARS-CoV-2 have not been well studied, the secondary structure prediction reveals that ORF 6 and 8 are transmembrane proteins and may have related functions as in SARS-CoV. In fact, the SARS-CoV-2 contains a full-length ORF 8, which in SARS-CoV this reading frame is divided into ORF 8a and ORF 8b. Linking of ORF 8a and ORF 8b into a single continuous gene fragment had no significant effect on virus growth and RNA replication in vitro [20], which indicates that there are ORFs of SARS-CoV-2 that may be similar to ORFs of SARS-CoV in function.

The ORF 6 does not induce apoptosis, nor inhibit cell gene expression, but inhibits the synthesis of IFN-β [25]. Due to the importance of ORF 6 and ORF 8, the roles of these subgenomic fragments and the encoded proteins in evading innate immunity will be analyzed in detail. At the early stage of the SARS epidemic, there is only an intact gene ORF8 encoding a single accessory protein in virus RNA [44]. At later stage, ORF 8 is split into two fragments ORF 8a and ORF 8b. Since ORF 8a in SARS-CoV is too small to function, ORF 8b is used as a comparison to ORF 8 in SARS-CoV-2. In addition, studies have shown that ORF 8b inhibits the IFN-β signaling pathway during virus infection, while protein ORF 8a does not show a similar effect [47]. We hypothesize that the HGS differences of the three ORFs may cause changes in the compatibility between virus and host, which can help explain the characteristics of SARS-CoV-2, such as rapid spread and unusual incubation period.

ORF 6 suppress innate immunity by blocking IFN signaling pathways (IFNAR)-(Tyk2+JAK1)-(STAT1+STAT2+IRF9)-(ISGs) (Figure 10). As a membrane protein with 63 amino acids, ORF 6 recruits karyopherin KPNA2 and KPNA1 through C-terminal tail toward the cytoplasm and bound them on ER/Golgi membrane [25]. KPNB1 is an essential factor helping transporting STAT1+STAT2+IRF9 complex into nucleus for ISRE activation in IFNα/β/γ signaling pathway (Figure 10) [27,30,32,52]. Thus, ORF 6 blocks the IFNAR-STAT signaling pathway by limiting the mobility of importin subunit KPNB1 and preventing STAT1 complex from moving into nucleus. Laboratory studies confirmed that expression of ORF 6 transform a sublethal infection into a lethal encephalitis and enhancing the growth of the virus in cells [39,93]. In addition, since KPNB1 is a commonly required protein by many nuclear import signaling pathways, ORF 6 plays a critical role in evading innate immune in vivo. In addition, ORF 6 can circumvent IFN production by inhibit IRF-3 phosphorylation in signaling pathway (TRAF3)-(TBK1+IKKi)-(IRF3)-(IFN-β), which is an essential part of signaling pathways triggered by viral sensors RIG-1/MDA5 and TLRs [94,95].

Protein ORF 8, sometimes named as ORF 8ab, is encoded by the single continuous ORF 8 (or ORF 8ab) in strains isolated at early stage SARS-CoV epidemic. The ORF 8b suppresses IFN signaling pathway by interacting with IRF3. It is reported that overexpression of ORF 8b and ORF 8ab brings significant effect on IRF3 dimerization, rather than IRF3 phosphorylation [47]. The 8b region of SARS-CoV protein ORF 8 has functions of ubiquitination binding, ubiquitination and glycosylation, which may interact with IRF 3 [95]. The expression of 8b and 8ab enhance the IRF 3 degradation, thus regulating immune functions of IRF 3. Interestingly, ORF 8 is an IFN antagonist expressed in the later stage of SARS-CoV infection. Studies show that activation of IRF 3 is blocked in the late stage of SARS-CoV infection, which was consistent with the late expression of 8b. Therefore, the expression of ORF 8 helps to suppress the innate immune response that may occur in the later stages of infection and delay the IFN-β signaling. This makes sense why the virus expresses a late stage IFN antagonist like ORF8. What other immune-modulating proteins will bind to ORF 8 still remain largely unknown and need further laboratory works.

The discovery of increased HGS of ORF 6 and ORF 8 provide a strong evidence that SARS-CoV-2 evolved to be more adaptable to humans than SARS-CoV. Based on these findings, following conjecture is proposed: that the SARS-CoV-2 genes involved in suppressing the host’s innate immunity are more powerful. Therefore, SARS-CoV-2 causes the delayed response of host innate immunity, which results in rapid transmission, low mortality and asymptomatic infection. These inferences are based on bioinformatics data, but offer a valuable picture of how SARS-CoV-2 could become different from SARS-CoV. In addition, the HGS method can also identify genetic mutations that help the virus adapt to humans.

### 3.4. The High HGS Mutation (ACT > ATT) Is Shared by Alpha, Delta and Omicron Variants

SARS-CoV-2 has high mutation rate. Several viral variants, including Alpha, Delta, and Omicron, have emerged in the past 2 years, but what kind of mutations contribute to viral adaptions is still not clear. Here, the top seven mutations in strains with the highest HGS were identified for Alpha, Delta, and Omicron variants with geolocation set as Europe. It is notable that the HGS of Delta variants is higher than others (Figure 3). Understanding how mutations help SARS-CoV-2 populations evolve under selection pressures is an important question. It is an effective method to analyze the distribution of various mutations among different variants.

Figure 11 shows the distribution of different mutations among the Alpha, Delta, and Omicron variants. These mutations are in the genomes with the top seven HGS. For genomes of Alpha variant, the mutations include ORF 8 mutation A202T (AAA > TAA), S mutation C185T (GAC > GAT) and T83C (AAT > AAC), N mutation G61-C (CGA > CCA) and C460T (AAC > AAT), M mutation C123T (ACT > ATT) and ORF 3a mutation C41T (ACA > ATA). For genomes of the Delta variant, the conserved mutants include ORF 6 mutations T29I (ACT > ATT), ORF 1ab mutation D143G > C (GAT > GGT), ORF 7a mutation A133V (GCT > GTT), M mutation Q15M (CAG > CTG), ORF 3a P47L (CCA > CTA), etc. 

Comparison of conserved mutations shows that genomes in three variants have both fully shared mutations and pairs shared mutations (Figure 11). Of all the gene mutations, the ACT > ATT mutation survived in high-HGS virus populations in all the three variants. It is worth noting that the high HGS mutations among the three variants are quite different. Only ACT > ATT is shared by all the variants. Five of the seven variants were specific to the Alpha variant. The same is true of the Delta variant. This finding strongly suggests that only very few mutations survived selection events and resulted in a new population of SARS-CoV-2 with high HGS, which could be more adapted to humans. In addition, the simultaneous occurrence of the mutation ACT > ATT in all three variants suggests that the mutation is highly stable in human-adapted strains. The underlying mechanisms of what type of mutations will be conserved in the SARS-CoV-2 remains poorly understood.

## 4. Conclusions

In this study, HGS is proposed to quantitatively analyze the genome similarity between virus and host. By applying the HGS analysis to SARS-CoV-2 variants, it is found that:There is correlation between the HGS and the inhibition of antivirus productions, which is a direct indicator of the virus to inhibit interferon synthesis.ORF 7b and ORF 8 of SARS-CoV-2 have increased HGS in the last 2 years, reaching 114% and 110% of the original value, respectively.By HGS and mutation analysis of genomes of Alpha, Delta, and Omicron variants, it is found that the mutation ACT > ATT is commonly shared in the high HGS strain population.

The significant increase in HGS distinguishes Delta from other variants. Delta variants may find out a way to improve host adaptation by selecting special gene mutations. The Omicron variant seems has relatively low HGS. The reason is still poorly understood. The lack of data may be a potential reason for this puzzle. Analysis of HGS of Omicron variants remains to be an interesting topic in the future. 

It is speculated that HGS may be helpful in explaining the quite different epidemiological characteristics of SARS-CoV-2, such with mild symptoms, rapid spread, and low mortality. However, the mechanism behind the impairment remains poorly understood and calls for future laboratory investigations. The Omicron variant appears to be less able to cause deaths than Alpha and Delta during the ongoing pandemic. However, there is still a serious warning sign about viral mutation. The threat of another coronavirus outbreak with high infectiousness and mortality remains an alarming possibility.

## Figures and Tables

**Figure 1 biomolecules-12-00972-f001:**
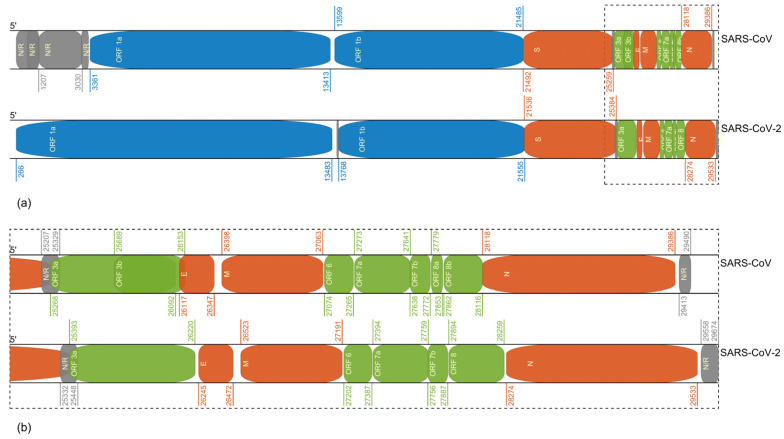
Open reading frames (ORF) organization in SARS-CoV-2 (GenBank: MN908947.3) and SARS-CoV (GenBank: AY394850.2) (**a**) The genome organization in complete sequence. (**b**) The enlarged diagram shows ORFs (no less than 75 nt in length) in SARS-CoV-2 and SARS-CoV genomes after 25,000 nt in sequence.

**Figure 2 biomolecules-12-00972-f002:**
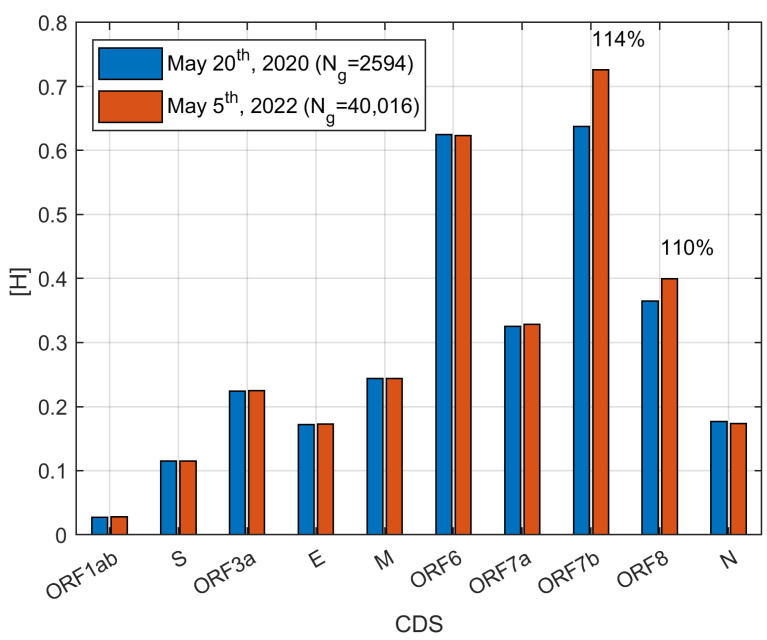
The mean HGS of ORF 7b and ORF 8 increased to 114% and 110% from 20 May 2020 to 5 May 2022. The HGS of other ORFs remained at almost the same level.

**Figure 3 biomolecules-12-00972-f003:**
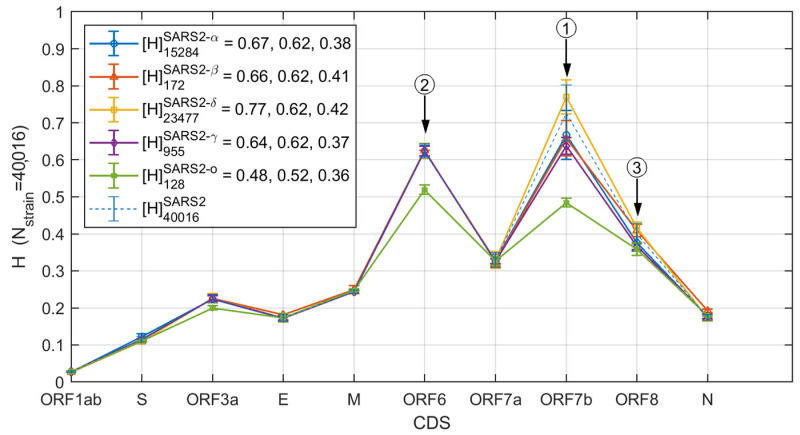
The SARS-CoV-2 Alpha, Beta, Delta, Gamma, and Omicron variants show significant difference in mean HGS values. The HGS values of the top 3 genes (ORF 7b, ORF 6 and ORF 8) are shown in legend for different variants.

**Figure 4 biomolecules-12-00972-f004:**
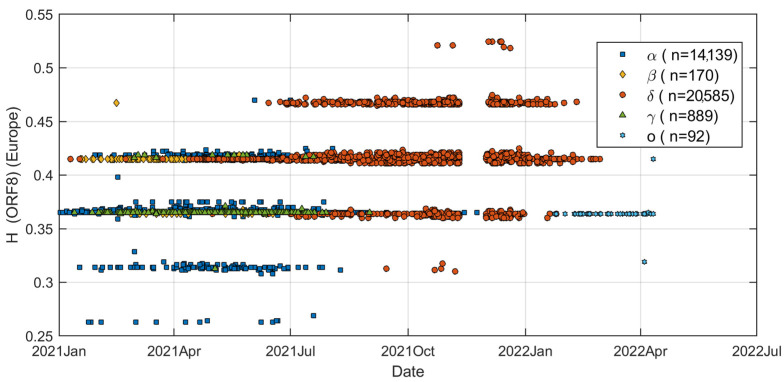
The average HGS value of ORF 8 gene in Delta variant is significantly higher than that in other variants. The Delta variant appeared between 2021 and 2022. The dataset contains 35,875 genomes with geolocation of Europe.

**Figure 5 biomolecules-12-00972-f005:**
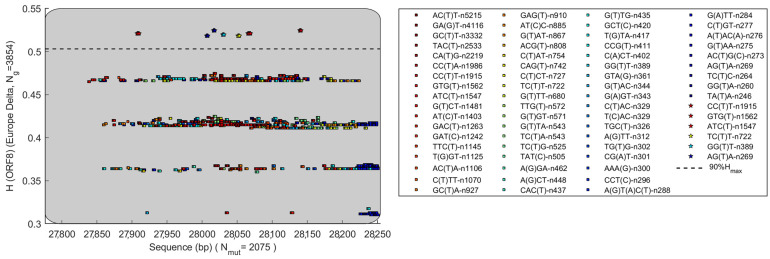
The ORF8 gene of Delta variant has mutated with different HGS. The dataset contains 3854 genomes with geolocation of Europe.

**Figure 6 biomolecules-12-00972-f006:**
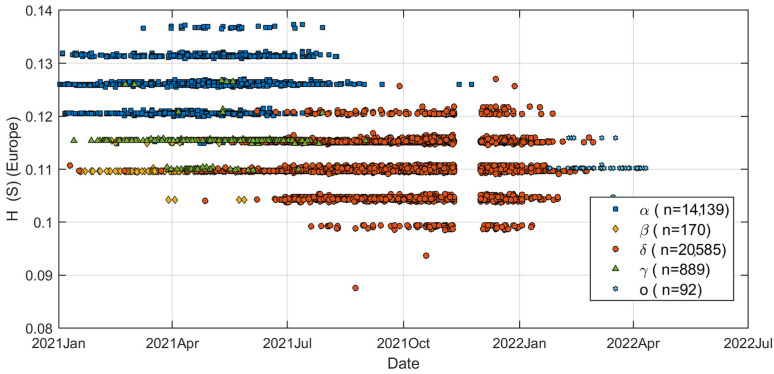
The average HGS of S gene in Delta variant is lower than that in Alpha variant. The Delta variant appeared between 2021 and 2022. The dataset contains 35,875 genomes with geolocation of Europe.

**Figure 7 biomolecules-12-00972-f007:**
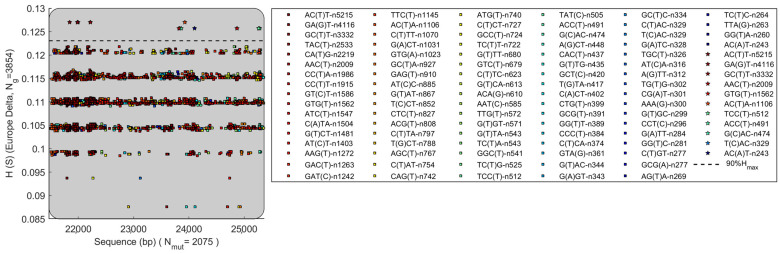
The ORF8 gene of Delta variant has mutated with different HGS. The dataset contains 3854 genomes with geolocation of Europe.

**Figure 8 biomolecules-12-00972-f008:**
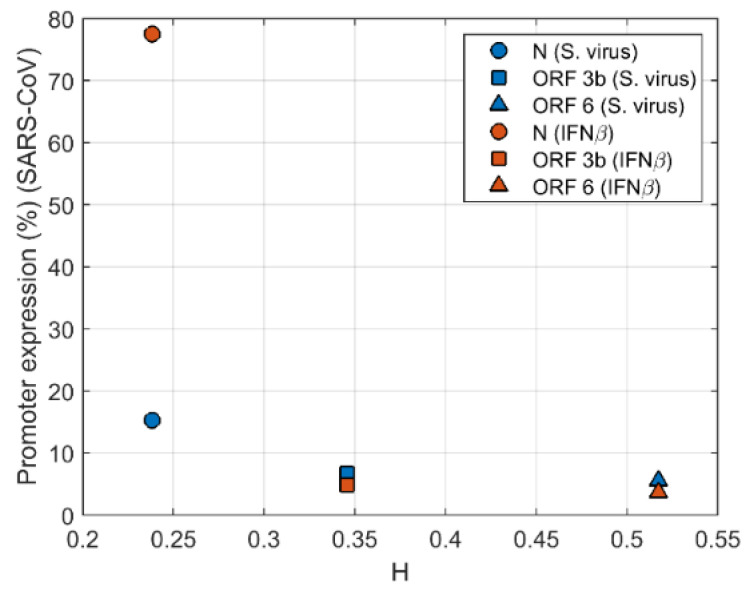
The SARS-CoV escapes innate immunity by suppressing ISRE (indicating interferon synthesis) promoter expression through ORFs encoding accessory proteins. The expression of ISRE-promoter decays rapidly (15.2%, 6.7% 5.6% for S. virus, 77.5%, 4.9%, 3.6% for IFN-β) along with the increasing HGS values.

**Figure 9 biomolecules-12-00972-f009:**
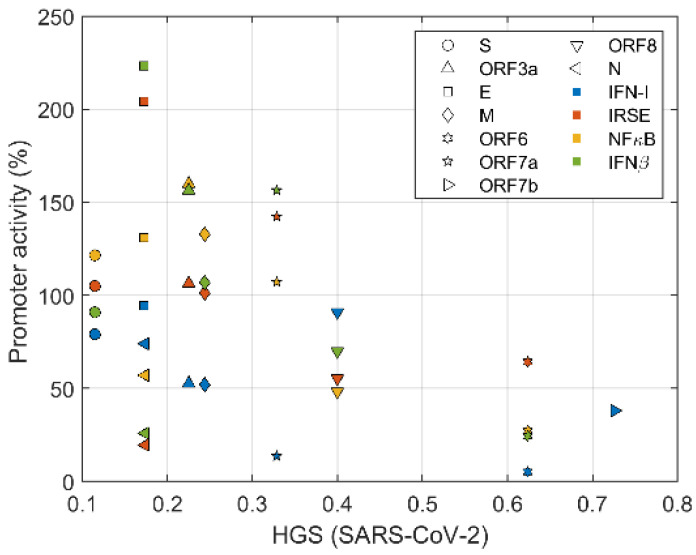
The SARS-CoV-2 escapes innate immunity by suppressing the expression of IFN-β, IFN-I, ISRE (indicating interferon synthesis) and NFκB through ORF encoding accessory proteins. As the HGS of viral genes increase, the activity of IFN-β, IFN-I, ISRE, and NFκB promoter decrease gradually.

**Figure 10 biomolecules-12-00972-f010:**
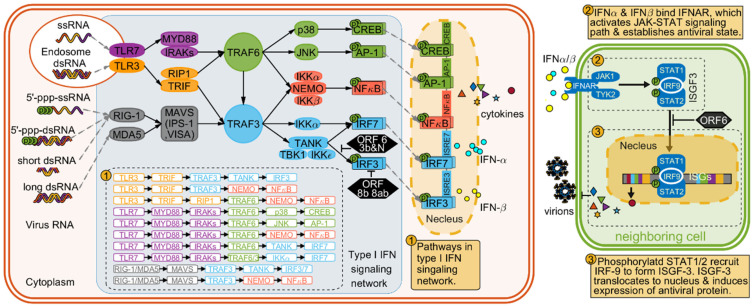
SARS-CoV induced immune response in host cells. Host cell detect virus invasion mainly by TLRs and RIG1-/MDA5 and lead to type I IFN signaling pathway. The receptor IFNAR senses type I IFN and leads to the JAK1-STAT signaling pathway, which expresses antiviral proteins and bring neighboring cell into anti-virus state. The ORF6 suppresses type I IFN expression by inhibiting translocation of STAT1+STAT2+IRF9 complex into nucleus. The ORF 6, 3b, N, and 8b/ab also inhibit the expression of type I IFN by blocking IRF3/7.

**Figure 11 biomolecules-12-00972-f011:**
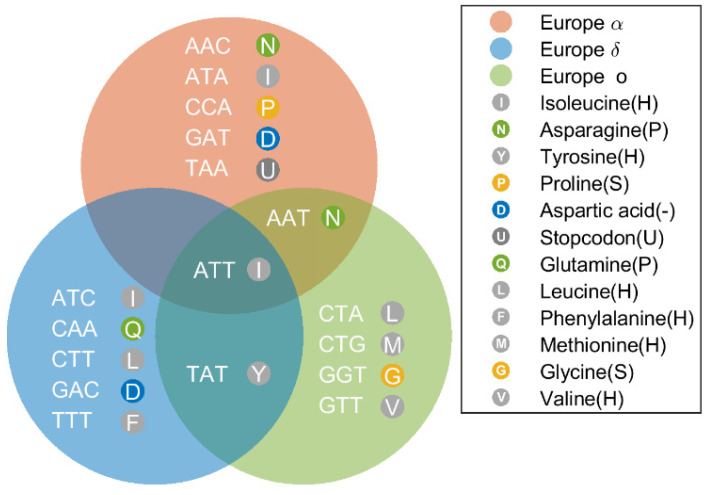
Highly conserved mutations identified in SARS-CoV-2 genomes from China, the USA and Europe. The mutation ATT(I) occurred in high-HGS populations in all the three variants.

**Table 1 biomolecules-12-00972-t001:** SARS-CoV-2 and SARS-CoV genome organization data. The ORF names defined in different papers are listed.

Narayanan	Marra	Rota	Start SARS	End SARS	Length SARS	Start SARS2	End SARS2	Length SARS2
ORF1a	ORF1a	1a	3361	13413	10053	266	13483	13218
N/R	N/R	N/R	13685	13759	75	13685	13759	75
ORF1b	ORF1b	1b	13398	21485	8088	13768	21555	7788
S	S-protein	S	21492	25259	3768	21536	25384	3849
N/R	N/R	N/R	25207	25329	123	25332	25448	117
ORF3a	ORF3	X1	25268	26092	825	25393	26220	828
E	E-protein	E	26117	26347	231	26245	26472	228
M	M-protein	M	26398	27063	666	26523	27191	669
ORF6	ORF7	X3	27074	27265	192	27202	27387	186
ORF7a	ORF8	X4	27273	27641	369	27394	27759	366
ORF7b	ORF9	N/R	27638	27772	135	27756	27887	132
ORF8a	ORF10	N/R	27779	27853	75	27894	28259	366
ORF8b	ORF11	X5	27862	28116	255	N/R	N/R	N/R
N	N-protein	N	28118	29386	1269	28274	29533	1260
N/R	N/R	N/R	29413	29490	78	29558	29674	117

## Data Availability

The HGS data for SARS-CoV-2 and SARS-CoV can be accessed at the website https://www.researchgate.net/profile/Weitao-Sun-4?ev=hdr_xprf.

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
