# Peer review of "Host-Genome Similarity Characterizes the Adaption of SARS-CoV-2 to Humans"

_biomolecules, 2022, doi:10.3390/biom12070972_

Round 1

Reviewer 1 Report

This paper proposes a quantitative analysis method to study the similarity between host genome and viral genome at specific open reading frames (HGS) that are related to the interaction with hosts. The study shows that SARS-CoV2 has significant increase of HGS in three ORFs than that of SARS-CoV. The findings about the difference in HGS among various types of SARS-CoV2 are interesting and they provide further suggestions for the relationship between HGS and the suppression of host immune activities. Below are some comments.

1.       Section 3 is really about Results and Discussions, rather than “Results”. It would be better if the writing distinguishes between results found in this article from discussion that are related to other earlier findings.

2.       The interpretation of the power law needs to be careful, since there are only three sample points. Figure 4 shows that H value can be used to distinguish between the on and off stage of the promotor expression.

3.       Figure 5 is helpful about the overall mechanism of innate host response. However, it is detached from the main results of the paper. Author may consider an effective way to make connection between the findings of this paper and the available biological knowledge.

“The mean HGS values of ORF6 and ORF8 in SARS-CoV-2 increased significantly, 168 reaching 122% and 148% of those of SARS-CoV ORF6 and ORF8, respectively 2.2 Viral 169 genome data” was under Section 2 Materials and Methods. Not sure what it is meant for.

Author Response

text of referee's comment:
1. Section 3 is really about Results and Discussions, rather than “Results”. It would be better if the writing distinguishes between results found in this article from discussion that are related to other earlier findings.

  1. The interpretation of the power law needs to be careful, since there are only three sample points. Figure 4 shows that H value can be used to distinguish between the on and off stage of the promotor expression.
  2. Figure 5 is helpful about the overall mechanism of innate host response. However, it is detached from the main results of the paper. Author may consider an effective way to make connection between the findings of this paper and the available biological knowledge.
  3. “The mean HGS values of ORF6 and ORF8 in SARS-CoV-2 increased significantly, 168 reaching 122% and 148% of those of SARS-CoV ORF6 and ORF8, respectively 2.2 Viral 169 genome data” was under Section 2 Materials and Methods. Not sure what it is meant for.

Answer:

  1. Thanks for the comments and suggestions. Section 3 contains the calculated HGS of 40016 SARS-CoV-2 genomes. Meanwhile, I analyze the HGS for different variants in this section, including Alpha, Delta, Omicron. Along with the findings that (1) ORF 7b, ORF 8 and ORF 6 are the top 3 genes with highest HGS, (2) ORF 7b and ORF 8 have obvious HGS increase in the last two years, discussion is given in this section on suppression of IFN-I signaling by these ORFs. The discussion is closely related the results in this section. So I revised the title of this section as “Results and discussions”
  2. Thanks for the suggestion. It’s true that the data of promoter expression is limited. Fitting a few data points to power law is not convincing. In the revised manuscript, I added more data to explain the correlation between expression of IFN-β, IFN-I, ISRE, NFκB and the HGS values. The data are following recent experiments on the inhibition of interferon by coronavirus proteins, including S, ORF3a, E, M, ORF6, ORF7a, ORF 7b, ORF8 and N[1,2]. The amount of data is sufficient to indicate a clear correlation between suppression of the immune system by viral proteins and their HGS values.
  3. Thanks for this comment. I moved Figure 5 and corresponding discussions to a new section “3.3 Suppression of IFN-I signaling by ORFs with different HGS values”. This new section is a discussion on the relation between the roles of ORFs in suppressing antivirus response and their high HGS values. This section explains how ORFs suppress the antivirus products and signaling paths. The new section is related to the main results for the reason that the key viral ORFs suppressing immune system have high HGS, and their HGS increases during last two years. Therefore, the results and discussion reveal that there is a relation between the HGS of ORF and its ability to suppress the immune system.
  4. Thanks for the comment. I remove this sentence in the revised manuscript.

Reference:

  1. Li, J.Y.; Liao, C.H.; Wang, Q.; Tan, Y.J.; Luo, R.; Qiu, Y.; Ge, X.Y. The ORF6, ORF8 and nucleocapsid proteins of SARS-CoV-2 inhibit type I interferon signaling pathway. Virus research 2020, 286, 198074, doi:10.1016/j.virusres.2020.198074.
  2. Xia, H.; Cao, Z.; Xie, X.; Zhang, X.; Chen, J.Y.; Wang, H.; Menachery, V.D.; Rajsbaum, R.; Shi, P.Y. Evasion of Type I Interferon by SARS-CoV-2. Cell reports 2020, 33, 108234, doi:10.1016/j.celrep.2020.108234.

Reviewer 2 Report

The manuscript describes a method to calculate the host-genome-similarity (HGS) of SARS-CoV and SARS-CoV2. The manuscript argues that the higher HGS of SARS-CoV-2 of open reading frame ORF8 ORF 7b and ORF 6 may be related to the suppression of innate immunity.

The HGS differences in Figure 2 are not very clear as there is a lack of statistically significant proof.  There are only 25 SARS and there are 30034 SARS2, only comparing the mean value is not convincing.  Also, the rationale for comparing SARS-CoV and SARS-CoV2 is lacking. They are related but it is not ready to assume that CoV2 is coming from CoV and it evolved to get higher HGS.

The manuscript also draws a power law behavior in Figure 4 and concludes that the HGS may be an indicator of the suppression of innate immunity. However, the power law extracted from only 3 unrelated ORFs are not reasonable. It would be more convincing if the same ORF has mutated with different HGS and got different expressions.  Also, anyone can fit a line with just 3 points to an equation with R =1, it is not easy to believe that there is a power law, it can be a linear law with outliers or something else.

Other questions/suggestions:

The parameters used in Blastn can change the results a lot. The author should reason the selection of the parameters. Using different parameters may get different answers?

On page 11 line 391, the ‘TLPs’ should be ‘TLRs’.

Point 3.2 ‘virus genes with high HGS are critical in suppressing innate immune’ is more proper to be ‘Virus genes related in suppressing innate immune tend to have high HGS’

In 3rd and 4th paragraphs in page 12, line 442 to line 470, the content is the pathway description of ORF6 and ORF8, which is not directly related to the conclusion or should be placed in a different place.

Author Response

text of referee's comment:
1. The HGS differences in Figure 2 are not very clear as there is a lack of statistically significant proof.  There are only 25 SARS and there are 30034 SARS2, only comparing the mean value is not convincing.  Also, the rationale for comparing SARS-CoV and SARS-CoV2 is lacking. They are related but it is not ready to assume that CoV2 is coming from CoV and it evolved to get higher HGS.

  1. The manuscript also draws a power law behavior in Figure 4 and concludes that the HGS may be an indicator of the suppression of innate immunity. However, the power law extracted from only 3 unrelated ORFs are not reasonable. It would be more convincing if the same ORF has mutated with different HGS and got different expressions.  Also, anyone can fit a line with just 3 points to an equation with R =1, it is not easy to believe that there is a power law, it can be a linear law with outliers or something else.

3.The parameters used in Blastn can change the results a lot. The author should reason the selection of the parameters. Using different parameters may get different answers?

  1. On page 11 line 391, the ‘TLPs’ should be ‘TLRs’.
  2. Point 3.2 ‘virus genes with high HGS are critical in suppressing innate immune’ is more proper to be ‘Virus genes related in suppressing innate immune tend to have high HGS’
  3. In 3rdand 4th paragraphs in page 12, line 442 to line 470, the content is the pathway description of ORF6 and ORF8, which is not directly related to the conclusion or should be placed in a different place.

Answer:

  1. Thanks for the comments and suggestions. I removed Figure 2 in the revised manuscript. It’s true that the amount of SARS-CoV genomes is limited and the comparison between HGS of SARS-CoV and SARS-CoV-2 may be not convincing. I remove the comparison between HGS of SARS-CoV and SARS-CoV-2.
  2. Thanks for the suggestion. It’s true that the data of promoter expression in Figure 4 is not enough. Fitting a few data points to power law is not convincing. In the revised manuscript, I added more data to explain the correlation between expression of IFN-β, IFN-I, ISRE, NFκB and the HGS values. The data are following recent experiments on the inhibition of interferon by coronavirus proteins, including S, ORF3a, E, M, ORF6, ORF7a, ORF 7b, ORF8 and N[1,2]. The amount of data is sufficient to indicate a clear correlation between suppression of the immune system by viral proteins and their HGS values. I also added a part on relation between the same ORF (ORF 8 and S, respectively) with various mutations and its HGS.
  3. Thanks for the comment and suggestion on Blastn parameters. I added the reason for choosing the parameter values in the revised manuscript.
    1. The expect threshold is a parameter that specifies the statistical significance threshold for reporting matches against database sequences. It describes how often an alignment with a given score is expected to occur at random. The default value (10) means that 10 such matches are expected to be found merely by chance, according to the stochastic model of Karlin and Altschul (1990). Lower EXPECT thresholds are more stringent, leading to fewer chance matches being reported. Here we used the default value 10 in matching the viral and host genomes.
    2. The choice of word size depends on balancing sensitivity and specificity. For nucleotide-nucleotide searches (i.e., "blastn") an exact match of the entire word is required before an extension is initiated. The minimum word size for blastn is 7. The default word size is 11. Here we used the default value 11 in matching genome segments.
    3. BLASTN uses a simple approach to score alignments, with identically matching bases assigned a reward and mismatching bases assigned a penalty. It is important to choose (absolute) reward/penalty ratio increasing for more divergent sequences. A ratio of 0.33 (1/-3) is appropriate for sequences that are about 99% conserved. Here we used a reward/penalty ratio of 0.66 (2/-3) for matching virus and host genome. Penalty for a nucleotide mismatch is -3 and reward for a nucleotide match is 2.
    4. A gap cost includes a value to open the gap and a value to extend the gap by a base. The default gap costs for other tasks supported by the blastn application is 5 to open a gap and 2 to extend one base. The choice of gap costs depends on the size of the expected gap. For simulating sequencing errors, the gap costs should be uniform and relatively high. Increasing the Gap Costs will result in alignments which decrease the number of Gaps introduced. Here we used the default values, i.e., gap costs are chosen as existence=5 and extension=2.
    5. Max matches in a query range Limit the number of matches to a query range. This parameter is useful if many strong matches to one part of a query may prevent BLAST from presenting weaker matches to another part of the query. Here I used Max matches as 1 in matching virus and host genome segment. It means that there are only two statuses in matching the virus and host genome segment: matched or not. If the virus segment matches the host genome (No matter how many times it matches), it only need to report once.
  4. Thanks for the comment. I revised the print error. this sentence in the revised manuscript.
  5. Thanks for the suggestion. I revised the section title ‘virus genes with high HGS are critical in suppressing innate immune’ to ‘Virus genes related in suppressing innate immune tend to have high HGS’.
  6. Thanks for this suggestion. I moved the content on the pathway description of ORF6 and ORF8 to a new section “3.3 Suppression of IFN-I signaling by ORFs with different HGS values”. This new section is a discussion on the relation between the roles of ORFs in suppressing antivirus response and their high HGS values. This section explains how ORFs suppress the antivirus products and signaling paths. The new section is related to the main results for the reason that the key viral ORFs suppressing immune system tend to have high HGS, and their HGS increase during last two years.

Reference:

  1. Li, J.Y.; Liao, C.H.; Wang, Q.; Tan, Y.J.; Luo, R.; Qiu, Y.; Ge, X.Y. The ORF6, ORF8 and nucleocapsid proteins of SARS-CoV-2 inhibit type I interferon signaling pathway. Virus research 2020, 286, 198074, doi:10.1016/j.virusres.2020.198074.
  2. Xia, H.; Cao, Z.; Xie, X.; Zhang, X.; Chen, J.Y.; Wang, H.; Menachery, V.D.; Rajsbaum, R.; Shi, P.Y. Evasion of Type I Interferon by SARS-CoV-2. Cell reports 2020, 33, 108234, doi:10.1016/j.celrep.2020.108234.

Round 2

Reviewer 2 Report

The author answers all my questions. 

If possible, I would suggest the author remove the power law in new figure 8.

Author Response

Response to Referee #2:

text of referee's comment:
The author answers all my questions. 

If possible, I would suggest the author remove the power law in new figure 8.

Answer

  1. Thanks for the suggestions. The power law lines have been removed in figure 8. And I also removed the sentences about power law fitting in the main text.